# Maternal Exposure to Ambient Air Pollution and Pregnancy Complications in Victoria, Australia

**DOI:** 10.3390/ijerph17072572

**Published:** 2020-04-09

**Authors:** Shannon M. Melody, Karen Wills, Luke D. Knibbs, Jane Ford, Alison Venn, Fay Johnston

**Affiliations:** 1Menzies Institute for Medical Research, University of Tasmania, Private Bag 23, Hobart, TAS 7001, Australia; karen.wills@utas.edu.au (K.W.); alison.venn@utas.edu.au (A.V.); fay.johnston@utas.edu.au (F.J.); 2School of Public Health, The University of Queensland, Herston, QLD 4006, Australia; l.knibbs@uq.edu.au; 3Clinical and Population Perinatal Health Research, Kolling Institute, Northern Sydney Local Health District, St Leonards, NSW 2065, Australia; Jane.ford@sydney.edu.au

**Keywords:** air pollution, pregnancy, gestational diabetes mellitus, preeclampsia, placental abruption

## Abstract

The relationship between maternal exposure to ambient air pollution and pregnancy complications is not well characterized. We aimed to explore the relationship between maternal exposure to ambient nitrogen dioxide (NO_2_) and fine particulate matter (PM_2.5_) and hypertensive disorders of pregnancy, gestational diabetes mellitus (GDM) and placental abruption. Using administrative data, we defined a state-wide cohort of singleton pregnancies born between 1 March 2012 and 31 December 2015 in Victoria, Australia. Annual average NO_2_ and PM_2.5_ was assigned to maternal residence at the time of birth. 285,594 singleton pregnancies were included. An IQR increase in NO_2_ (3.9 ppb) was associated with reduced likelihood of hypertensive disorders of pregnancy (RR 0.89; 95%CI 0.86, 0.91), GDM (RR 0.92; 95%CI 0.90, 0.94) and placental abruption (RR 0.81; 95%CI 0.69, 0.95). Mixed observations and smaller effect sizes were observed for IQR increases in PM_2.5_ (1.3 µg/m^3^) and pregnancy complications; reduced likelihood of hypertensive disorders of pregnancy (RR 0.95; 95%CI 0.93, 0.97), increased likelihood of GDM (RR 1.02; 95%CI 1.00, 1.03) and no relationship for placental abruption. In this exploratory study using an annual metric of exposure, findings were largely inconsistent with a priori expectations and further research involving temporally resolved exposure estimates are required.

## 1. Introduction

Ambient air pollution is the leading environmental risk factor contributing to global disease burden. Associations between air pollution and respiratory and cardiovascular disease are well characterized [1]. More recently, pregnant women have been identified as a subgroup of the population who are particularly vulnerable to the health harms associated with air pollution, however this is largely due to emerging evidence concerning birth outcomes observed in the neonate associated with maternal exposure to air pollution [2]. Somewhat less is understood regarding potential impacts of maternal exposure to air pollution during pregnancy and pregnancy conditions such as pregnancy-induced hypertension, preeclampsia, gestational diabetes mellitus (GDM) and placental abruption.

The same direct and indirect effects of maternal ambient air pollution exposure that have been postulated to explain observed adverse fetal growth and maturity outcomes, may also underpin pregnancy complications. Direct mechanisms have been demonstrated in a number of animal studies and include particle translocation to the placenta, which triggers local inflammation [3]. More recently, translocation of black carbon particles following maternal exposure in pregnancy has been demonstrated on the fetal side of human placentae [4]. Indirectly, exposure to ambient air pollutants have been demonstrated to trigger oxidative stress and systemic inflammation, which may in turn affect placental growth, development and function [5].

Pregnancy conditions, such as hypertensive disorders and GDM, are associated with significant short- and longer-term morbidity and mortality for both mother and child. For example, preeclampsia is a major cause of maternal morbidity through seizures, stroke and liver rupture and adverse perinatal outcomes, such as prematurity and intrauterine growth restriction [6]. GDM is associated with proximate adverse outcomes such as neonatal hypoglycaemia and birth trauma, and more distal outcomes such as increased risk of cardiometabolic disease in later life for both mother and child [7]. Placental abruption, the premature separation of the placenta prior to delivery, is a life-threatening condition for the fetus and mother, associated with maternal haemorrhage and secondary consequences such as hypovolemic shock, renal failure, disseminated intravascular coagulation, emergency hysterectomy and perinatal and maternal mortality [8].

Existing studies exploring an association between maternal exposure to ambient air pollution and pregnancy complications, such as hypertensive disorders of pregnancy, GDM and placental abruption, are few. A meta-analysis found both nitrogen dioxide (NO_2_) and fine particulate matter (PM_2.5_) were associated with an increased risk of pregnancy-induced hypertensive disorders, with PM_2.5_ demonstrating a relatively stronger association than NO_2_ [9]. Evidence of an adverse association between air pollution and GDM is less well-established. A meta-analysis of 11 studies exploring NO_2_, nitrogen oxides, ozone and sulfur dioxide only found an association between second trimester PM_2.5_ and first and second trimester NO_x_ exposure and GDM [10]. Only a handful of studies have specifically explored ambient air pollution and placental abruption; with all three studies finding an adverse association [11,12,13]. Synthesising findings is challenging owing to the relatively limited evidence base, various pollutants studied, various ranges of exposure (high-level vs. low-level), various periods of exposure (whole of pregnancy, trimester-specific or days prior to labour) and definition of outcomes. Our analysis aimed to examine a state-wide cohort of pregnancies in Victoria, Australia to evaluate possible associations with maternal exposure to low-level ambient NO_2_ and fine particulate matter (particulate matter with an aerodynamic diameter less than 2.5 micrometres; PM_2.5_) over the course of a pregnancy on the likelihood of hypertensive disorders of pregnancy, GDM and placental abruption.

## 2. Materials and Methods

Ethics approval for this study was obtained from the Tasmania Health and Medical Human Research Ethics Committee (ref H0015033).

### 2.1. Data

#### 2.1.1. The Study Population

We defined our cohort using pregnancy and delivery data of singleton births of mothers resident in the state of Victoria and delivery from 1 March 2012 to 31 December 2015 that were recorded in the Victorian Perinatal Data Collection (VPDC). Victoria is the second most populous state in Australia (population 5,926,624) [14]. The VPDC is a statutory collection of information on all births occurring in Victoria, Australia. Data on maternal characteristics, pregnancy care, birth and neonatal outcomes are collected for all births ≥20 weeks gestation or at least 400 g if the gestation is unknown, usually by a midwife. Validation against medical records indicates positive predictive values of 92% on reporting for pre-eclampsia and 94% for GDM [15].

#### 2.1.2. Outcome Assessment

The outcomes of interest were hypertensive disorders of pregnancy (pregnancy-induced hypertension, pre-eclampsia and eclampsia), GDM and placental abruption. For all conditions, timing of diagnosis during pregnancy was not known, simply the presence or absence of a condition as recorded at birth. Hypertensive disorders of pregnancy: Blood pressure measurement and urinalysis are performed routinely at most antenatal visits, and pregnancy-induced hypertension, preeclampsia and eclampsia are diagnosed according to the International Classification of Diseases 10th Revision (ICD-10AM) codes O13-O15. Pre-existing hypertensive disorders were not included. *GDM:* In Australia, it is recommended that pregnant women are screened for GDM between 24 and 28 weeks gestation with an oral glucose tolerance test. Women who are recognised as high-risk may be screened earlier in pregnancy. Current guidelines recommend a diagnosis of GDM if the 2 h oral glucose tolerance test result is greater than 8.5 mmol/L following a 75 g oral glucose load [16]. The diagnostic threshold was lowered during the study period to align with the WHO-2013 diagnostic criteria, with recommendation from the Royal Australian and New Zealand College of Obstetricians and Gynaecologists that the new criteria be adopted by 1 January 2015. Previous diagnostic thresholds included a 2 h glucose level of greater than or equal to 11.1 mmol/L following a 75 g oral glucose load, a random plasma glucose of greater than or equal to 11.1 mmol/L or a fasting glucose of greater than or equal to 7.0 mmol/L. The lowering of the diagnostic threshold means that the prevalence of GDM in the Australian population would have increased during the study period [17]. We have adjusted for this by including year of conception in the statistical models. If a diagnosis of GDM did not appear in the VPDC (ICD-10AM code O24.4), it was presumed women did not have GDM. Pre-existing diabetes mellitus cases were not included. Placental abruption: Placental abruption is typically a clinical diagnosis characterised by either an acute process of excessive or premature placental detachment or a chronic process representing inadequate placental attachment. Abruption is diagnosed according to ICD-10 AM codes O45.

#### 2.1.3. Exposure Data

There were two primary exposures of interest; estimated annual NO_2_ and PM_2.5_ concentrations. In Australia, PM_2.5_ is derived from many sources including dust, sea salt and combustion emissions (e.g., vehicles, landscape fires, coal-fired power generation), NO_2_ is more reflective of traffic-related and, to a less extent, industrial sources of air pollution [18].

Annual NO_2_ concentration: We estimated annual average NO_2_ at the centroid of each census mesh block. The mesh block is the smallest geographical area defined by the Australian Bureau of Statistics and form larger geographical units as defined by the Australian Statistical Geography Standard. Mesh blocks contain an average of 30 to 60 dwellings [19]. Ambient average annual NO_2_ was estimated using a land-use regression (LUR) model, which included satellite observations of tropospheric NO_2_ columns with other spatial predictor variables. The model explained 81% (RMSE: 1.4 ppb; parts per billion) of the spatial variability in measured NO_2_, based on five-fold cross-validation, and, in an external validation, 66% of measured NO_2_ (RMSE: 2 ppb) at urban locations. The methods used to develop the LUR model is outlined in detail elsewhere [18].

Annual PM_2.5_ concentration: Annual ambient PM_2.5_ estimates at the mesh block level were also derived from a LUR model, which were derived by relating satellite-observed aerosol optical depth to ground-level PM_2.5_, and other predictors as described by Knibbs et al., 2018. The LUR model explained 52% (RMSE: 1.2 µg/m^3^) of spatial variability in measured annual PM_2.5_ in an external validation [20].

Assignment of exposure: Annual NO_2_ and PM_2.5_ exposure was assigned to either the single year of pregnancy or as a weighted average as months pregnant in year one and year two, if the pregnancy spanned two calendar years. Although annual PM_2.5_ and NO_2_ data was available at the mesh block level, exposure was assigned at the geographical unit of statistical area level 1 (SA1), as this was the geographical unit available for residential address at birth. Each SA1 contains approximately 400 individuals [21]. Because multiple mesh blocks form SA1s (range 1–19), a population-weighted average of NO_2_ and PM_2.5_ was calculated per mesh block, to inform NO_2_ and PM_2.5_ exposure per SA1. Population counts per mesh block were extracted using ‘persons usually resident’ data from the 2011 Australian Bureau of Statistics Census of Population and Housing [19].

#### 2.1.4. Weather and Other Covariates

We obtained minimum and maximum daily temperature data from the Australian Government Bureau of Meteorology for all stations in the state of Victoria (see Appendix A for a list of weather stations). The most geographically proximate weather station was allocated per SA1. The 24 h mean temperatures were calculated. We calculated whole of pregnancy averages from conception to delivery to adjust for underlying temporal and seasonal impacts. The estimated date of conception was calculated by subtracting 266 days from the estimated date of confinement.

#### 2.1.5. Confounding

We addressed potential confounding by considering a priori several maternal, pregnancy, meteorological and temporal characteristics. Maternal characteristics included mother’s age (continuous outcome and categorical outcome: ≤19 years, 20 to 34 years, ≥35 years old), area-level maternal socioeconomic position (Australian Bureau of Statistics Index of Relative Socioeconomic Disadvantage (IRSD) quartile assigned to maternal residence at the level of SA1) and remoteness classification (as per the Australian Bureau of Statistics Australian Statistical Geography Standard classification of major capital city, inner regional, outer regional, remote and very remote). IRSD is a composite measure that summarises information about economic and social circumstances of people and households within an area. Pregnancy-related characteristics included: parity (nulliparous vs. multiparous) and smoking in early pregnancy (first 20 weeks; yes/no). Meteorological characteristics included whole of pregnancy mean daily temperature (°C). Temporal characteristics included season of conception (winter, spring, summer and autumn) and year of conception.

We included singleton births in our analyses, as pregnancies characterised by multiple births are systematically different from singleton births. We were not able to distinguish multiple pregnancies within the study period to the same mother; therefore, eligible births may include more than one pregnancy to the same woman.

### 2.2. Statistical Analysis

Statistical analysis was conducted using R (v 3.4.0) [22]. The association between exposure to air pollution and outcomes (all dichotomous), were estimated using multivariable log-link binomial generalised linear models. Both pollutants were included in the log-binomial regression models. Missing data were handled using multiple imputation by chained equations (MICE package in R) [23]. Twenty imputed datasets were considered adequate. Covariates included in the multivariable regression models were used to impute the missing values. Regression models were fitted using the imputed data. Confounders were defined based on robust a priori evidence or if there was evidence that the covariate was associated with exposure, associated with the outcome and adjustment for the covariate altered the coefficient by greater than ten percent. We set the statistical significance level (α) at 0.05. Results are presented as the change in outcome per interquartile range (IQR) increase in each pollutant. Correlation between the two pollutants was assessed to determine the use of a co-pollutant model in the main analysis, and single pollutant model in the sensitivity analysis. Sensitivity analyses were conducted to examine the robustness of the main results, which included examining associations for single pollutant models, assigning exposure based on year of conception (rather than weighted across years of pregnancy), adjusting for remoteness, adjusting for fixed cohort bias, a subset analysis restricting to term births only (estimated gestational age ≥37 and ≤42 weeks) and a subset analysis specifically for GDM restricting to births occurring at 28 weeks and greater (when the majority of women would have had an opportunity for screening). Fixed cohort bias was assessed by limiting to include pregnancies with conception dates between <22 weeks before the cohort started and >42 weeks before the cohort ended, modeled on an approach outlined by Strand [24].

## 3. Results

There were 285,594 singleton births to women resident in Victoria between 1 March 2012 and 31 December 2015. Infant, maternal, pregnancy and labour characteristics and birth outcomes are outlined below in Table 1. The proportion of missing data was generally low. The most common missing variables included neonatal admission to special care nursery/neonatal intensive care unit (3.7% missing) and smoking in early pregnancy (1.5% missing). The remainder of variables were missing for <0.1% of data.

### 3.1. Exposure to Ambient Air Pollution

The average annual ambient NO_2_ exposure was 6.0 (ppb) (median 5.6; IQR 3.9). The average annual PM_2.5_ exposure was 6.9 µg/m^3^ (median 7.1; IQR1.3) (Figure 1). Ambient annual NO_2_ and PM_2.5_ were mapped to maternal residence at birth in Figure 2 (analysed at the SA1 level and presented for illustration at the SA2 level). These values are below the current annual ambient air quality standards for Australia of 30 ppb for NO_2_ and 8 µg/m^3^ for PM_2.5_, as outlined by the National Environment Protection (Ambient Air Quality) Measure [25]. Annual ambient NO_2_ was notably higher in areas classified as ‘Greater Melbourne’ compared with ‘Rest of Victoria’, and PM_2.5_ demonstrated more geographic variation, although was slightly higher in ‘Greater Melbourne’ compared with the rest of Victoria (as classified by the Greater Capital City Statistical Areas by the Australian Bureau of Statistics; Appendix A). The two pollutants were poorly correlated (r = 0.37).

### 3.2. Association Between Ambient NO_2_ and PM_2.5_ and Pregnancy Conditions

IQR increases in ambient NO_2_ and PM_2.5_ were associated with an 11% and 5% reduced likelihood of hypertensive disorders of pregnancy respectively, including pregnancy-induced hypertension, preeclampsia and eclampsia (NO_2_ RR 0.89; 95%CI 0.86, 0.91; *p* < 0.0001 per IQR increase; PM_2.5_ 0.95; 95%CI 0.93, 0.97; *p* < 0.0001 per 1QR increase). The direction of effect for GDM differed by pollutant, whereby an IQR increase in NO_2_ was associated with an 8% reduced likelihood and PM_2.5_ a 2% increased likelihood of GDM (NO_2_ RR 0.92; 95%CI 0.90, 0.94; *p* < 0.0001 per IQR increase; PM_2.5_ 1.02; 95%CI 1.00, 1.03; *p* < 0.031 per IQR increase). Ambient NO_2_ was associated with a 19% reduced likelihood of placental abruption (RR 0.81; 95%CI 0.69, 0.95; *p* < 0.011 per IQR increase) and no association was observed for ambient PM_2.5_ (RR 1.06; 95%CI 0.94, 1.20; *p* = 0.35 per IQR increase) (Table 2).

### 3.3. Sensitivity Analyses

We repeated the analyses where exposure was assigned to a single year, the year of conception. Findings were similar to those in the main analyses but tended to have smaller effect sizes closer to the null (Appendix A). We also fitted separate single-pollutant regression models for NO_2_ and PM_2.5_ to assess the independent associations of each pollutant with obstetric outcomes. Findings were similar to the main analyses; however, effect estimates were marginally further from the null using single pollutant models. Additionally, there was no independent association between ambient PM_2.5_ and increased likelihood of GDM (Appendix A). We did not have data pertaining to antenatal attendance or health service access, which may introduce ascertainment bias to these relationships. As access to health care services generally reduces with increasing remoteness, we conducted a sensitivity analysis introducing remoteness into the models as a proxy for antenatal care. Findings were very similar to the main analyses after adjusting for remoteness (Appendix A). We also conducted a subset analysis restricting to term births (estimated gestational age ≥37 and ≤42 weeks), to restrict to pregnancies that had comparable gestational lengths. We found no substantial differences in this subset analysis, although the adverse finding between ambient PM_2.5_ and GDM was no longer statistically significant (*p* = 0.054) (Appendix A). Adjusting for potential fixed cohort bias, whereby long pregnancies at the beginning of the cohort and short pregnancies at the end of the cohort are excluded, did not substantially alter the results (Appendix A). Results for a subset analysis exploring the association between pollutants and GDM in births equal to or greater than 28 weeks gestation, in order to restrict to pregnancies that would have had an ‘opportunity’ for GDM screening, did not alter the observed associations (results not shown).

## 4. Discussion

Our findings exploring pregnancy complications were largely inconsistent with a priori expectations framed from previous, albeit limited studies. Ambient NO_2_ exposure was associated with a reduced likelihood of all studied pregnancy conditions. Mixed findings were evident regarding ambient PM_2.5_ and the outcomes of interest, with reduced likelihood of hypertensive disorders, increased likelihood of GDM and no association for placental abruption observed. Adjusting analyses by assigning exposure based purely on year of conception, use of single pollutant models, adjusting for a proxy marker of health service access, and limiting the analysis to term births, did not meaningfully change the results.

At odds with our findings of a reduced likelihood of hypertensive disorders of pregnancy associated with exposure to both pollutants, a meta-analysis of ambient air pollution and hypertensive disorders of pregnancy demonstrated both NO_2_ and PM_2.5_ exposure were associated with an increased likelihood of pregnancy-induced hypertension and preeclampsia. The meta-analysis found 5 µg/m^3^ increases in PM_2.5_ were associated with 47% increase in pregnancy-induced hypertensive disorders (pooled OR 1.47; 95%CI 1.27, 1.68) and a 10 µg/m^3^ increase in NO_2_ associated with a 11% increase (pooled OR 1.11; 95%CI 1.00, 1.21) [9]. Noting that comparison of our findings with other studies is limited by our use of an annual average of exposure, as opposed to the more commonly used trimester-specific or whole of pregnancy estimates in comparable studies. Additionally, the exposure increments presented in the meta-analysis are approximately three times that of our exposure context. The reason our findings differ in the direction of association is unclear but may be explained by biases introduced from residual confounding, or exposure misclassification resulting from the use of an annual metric of exposure that may not capture important within-pregnancy variations. Factors which we were unable to control for, such as access to health care and engagement in antenatal services, may be contributing to ascertainment biases.

We found NO_2_ exposure reduced the likelihood, and PM_2.5_ increased the likelihood, of GDM. While there is substantial evidence concerning ambient air pollution and type 2 diabetes in the non-pregnant adult population, the association between ambient air pollution and GDM is less well characterised [26]. A meta-analysis of 11 studies exploring this relationship only found second trimester PM_2.5_ and first and second trimester NO_x_ exposure were associated with increased likelihood of GDM, with no association observed in pooled estimates for NO_2_, ozone or sulfur dioxide [10]. Again, comparison of our findings with those presented in the meta-analysis is challenging due to the higher exposure increments presented in the meta-analysis, highlighting the limited evidence in the low-level pollution setting, as well as the challenges in comparing heterogenous exposure periods (annual versus trimester-specific or whole of pregnancy). Discordance in the direction of associations observed between specific pollutants and various outcomes may reflect the differing composition and effect of individual pollutants. For example, NO_2_ is a proxy of traffic related air pollution, whereas fine particulate matter may arise from various sources, including geogenic dust, industrial activity and domestic wood heater use. Our findings are consistent with our earlier work which demonstrated an association of PM_2.5_ with a severe smoke event resulting from a coal mine fire in regional Victoria, Australia and increased likelihood of GDM [27]. To our knowledge, only three studies have explored maternal exposure to PM_2.5_ and NO_2_ in relation to placental abruption, with all reporting an increased likelihood of abruption with greater exposure [11,12,13].

The mechanisms by which maternal exposure to ambient air pollution are thought to cause maternal and/or fetal harm are not entirely clear, however potential proposed mechanisms include indirect (e.g., intrauterine inflammation) and/or direct mechanisms (e.g., placental translocation of particles) [4,28]. Accumulation of black carbon on the fetal side of the placenta following residential exposure to black carbon during pregnancy has been demonstrated and suggests a potential mechanism by which adverse pregnancy outcomes may occur [4]. Although such mechanisms would not explain our observed associations. There are many potential mechanistic pathways that have not yet been elucidated. There is increasing evidence that environmental toxicants, such as dioxins, alter placental microRNA expression, which are thought to play a role in placental-mediated conditions such as preeclampsia and fetal growth restriction [29,30,31]. The interaction between microRNAs and air pollution in humans is not known. Additionally, the role of other reproductive conditions, such as endometriosis, and environmental pollutants is unclear. There is in vitro evidence that the effect of dioxins on progesterone-mediated cannabinoid expression is dysregulated in women with endometriosis [32]. Beyond the direct maternal morbidity and mortality caused by our studied obstetric conditions, there is merit in further understanding these possible associations between ambient air pollution and pregnancy complications, as it may aid our understanding of the mechanistic pathway between ambient air pollution exposure and more distal neonatal outcomes, such as preterm birth and fetal growth restriction [2].

Strengths of this study include the use of routinely collected deidentified data, which enabled study of a complete cohort of all births within the state spanning almost four years. In doing so, we had a relatively large sample size of just under 300,000 births. Additionally, use of exposure metrics derived from validated satellite-based LUR models, enabled state-wide spatial coverage which was geographically resolved level of SA1, of which there are approximately 15,000. The SA1 is the smallest geographical unit made available by the data custodian. We acknowledge limitations of this study, including the use of an annual metric of exposure. More temporally refined exposure data was not available at the SA1 level. The use of an annual average of exposure provides a surrogate for a woman’s long-term background exposure. Perinatal epidemiology studies such as this typically utilise trimester-specific or whole of pregnancy exposure estimates [9,10,11,12,13]. Limited studies in this discipline have utitlised annual averages of pollutants, or proxies of pollutants (e.g., traffic density as a proxy of traffic-related air pollution) [33,34,35]. However, the use of an annual exposure metric is not uncommon for studies in the non-pregnant population, particularly for studies of allergic and respiratory disease [36,37,38]. This limits the comparability of our findings with others and further adds to the heterogenous nature of the evidence base. However, our findings would be most comparable to studies utilising a whole of pregnancy exposure estimate given the relative duration of these two estimates. Use of an annual exposure metric does not allow for exploration of trimester-specific associations of ambient air pollutants and outcomes of interest, or pre-conception exposures, or transient ‘spikes’ in air quality, which may be important. Additionally, we are unable to explore relationships for PM_2.5_ by composition or likely source, such as PM_2.5_ arising from geogenic dust versus industrial activities.

We were also unable to explore the temporality of observed relationships, as data were limited to the presence or absence of a condition rather than the timing of diagnosis in pregnancy. It is possible that despite attempting to control for a range of important maternal, pregnancy, meteorological and temporal covariates, that residual confounding may bias the results. For example, as both exposure and outcome will be heavily influenced by maternal socioeconomic conditions, if the area-level IRSD score does not completely capture this, then residual confounding may bias our findings and the direction of potential bias is unclear. Other individual-level markers of socioeconomic status were unavailable in the dataset (e.g., maternal education, income level, employment etc.). Additionally, we were unable to adjust for some potentially important covariates, including maternal stress, maternal anthropometrics such as body mass index, maternal alcohol consumption, antenatal attendance, health service access and health literacy, among others. Obstetric care may differ substantially between public and private models of care, and we were unable to account for health insurance status in our analysis. Possible effect modifiers of the examined relationships, such as proximity to major roads, traffic density and green space, were beyond the scope of this study. Use of fixed maternal residential address at time of delivery is common practice in environmental epidemiology studies such as ours. However, it may introduce exposure misclassification by not accounting for maternal mobility in pregnancy, such as movements to areas with different pollutant profiles for work, time spent indoors or outdoors, or indoor air quality. The importance of residential mobility during pregnancy in introducing bias remains unclear. In a previous study of ambient air pollution, there was negligible change in outcomes in accounting for residential mobility [39]. Despite these limitations, these exploratory findings provide a useful insight into possible associations between background air pollution exposure and obstetric complications.

## 5. Conclusions

In this exploratory study of a large state-wide cohort of births in Victoria, Australia, maternal exposure to low-level annual ambient NO_2_ was associated with a reduced likelihood of hypertensive disorders, GDM and placental abruption. Ambient PM_2.5_ exposure was associated with reduced likelihood of hypertensive disorders of pregnancy, increased likelihood of GDM and not associated with placental abruption. Our findings were largely inconsistent with a priori expectations, although there are relatively few studies exploring these relationships to frame our expectations. Our findings may reflect the true nature of these relationships or may have resulted from biases introduced by limitations in our exposure, outcome or covariate data. Further research using more temporally resolved estimates of both exposure and outcome is needed to elucidate the nature of these relationships.

## Figures and Tables

**Figure 1 ijerph-17-02572-f001:**
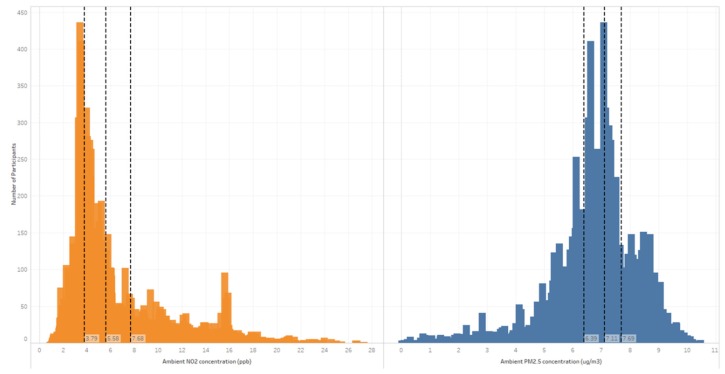
Ambient NO_2_ (ppb) and PM_2.5_ (µg/m^3^) for the study population assigned to maternal residence at time of delivery.

**Figure 2 ijerph-17-02572-f002:**
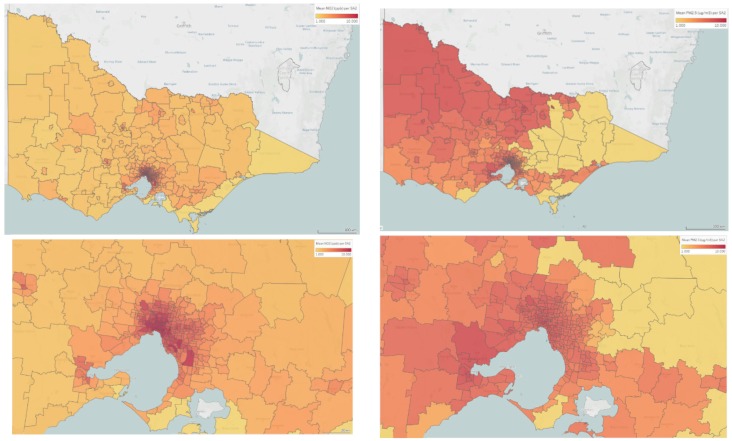
Annual maternal ambient air pollution exposure assigned to maternal residence at time of birth mapped to SA2 level for mean ambient annual NO_2_ (ppb) (left) and for mean ambient annual PM_2.5_ (µg/m^3^) (right) for the state of Victoria (top) and Greater Melbourne (bottom).

**Table 1 ijerph-17-02572-t001:** Description of all singleton births in Victoria 1 March 2012 to 31 December 2015 (*n* = 285,594).

Infant Characteristics	*n* (%)
Female gender	138,979 (48.7)
Aboriginal and/or Torres Strait Islander	4291 (1.5)
Admitted to special care nursery	36,930 (12.9)
Admitted to neonatal intensive care unit	3890 (1.4)
Liveborn	284,332 (99.6)
Maternal characteristics	
Maternal smoking in early pregnancy (<20 weeks)	28,283 (9.9)
Maternal smoking in late pregnancy (>20 weeks)	16,340 (5.7)
Country of birth Australia	183,441 (64.2)
Aboriginal and/or Torres Strait Islander	3587 (1.3)
Resident in major capital city	166,285 (58.2)
Pregnancy and labour characteristics	
Nulliparous	127,501 (44.6)
Spontaneous onset of labour *	101,725 (35.6)
Caesarean section birth	93,121 (32.6)
Hypertensive disorder of pregnancy ^	9987 (3.5)
Gestational diabetes mellitus	23,035 (8.1)
Placenta praevia	1277 (0.5)
Placental abruption	399 (0.1)
Year of birth	
2012	62,018 (21.7)
2013	73,909 (25.9)
2014	74,802 (26.2)
2015	74,865 (26.2)

* Spontaneous birth is defined as those where labour type was not coded as ‘induced’ or ‘no labour’. ^ inclusive of pregnancy-induced hypertension, preeclampsia, eclampsia.

**Table 2 ijerph-17-02572-t002:** Association between average ambient NO_2_ and PM_2.5_ exposure in pregnancy and selected pregnancy conditions for births in Victoria, Australia between 1 March 2012 and 31 Dec 2015.

Pregnancy Condition	Adjusted Relative Risk (95%CI); *p* Value	
	Per IQR Increase in Annual Ambient NO_2_ Concentration (ppb)	Per IQR Increase in Annual Ambient PM_2.5_ Concentration (µg/m^3^)
Hypertensive disorder of pregnancy (pregnancy-induced hypertension, preeclampsia, eclampsia) *	**0.89 (0.86, 0.91); <0.0001**	**0.95 (0.93, 0.97); <0.0001**
Gestational Diabetes Mellitus **	**0.92 (0.90, 0.94); <0.0001**	**1.02 (1.00, 1.03); 0.03**
Placental abruption **	**0.81 (0.69, 0.95); 0.01**	1.06 (0.94, 1.20); 0.35

* Adjusted for maternal age ≥35, parity, Index of Relative Socioeconomic Disadvantage, average ambient temperature over whole of pregnancy. ** Adjusted for smoking in early pregnancy, maternal age (years), parity, Index of Relative Socioeconomic Disadvantage, year of conception, average ambient temperature over whole of pregnancy. Results for co-pollutant models presented. Bold typeface indicates statistically significant *p* < 0.05.

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
