# Peer review of "Maternal Exposure to Ambient Air Pollution and Pregnancy Complications in Victoria, Australia"

_ijerph, 2020, doi:10.3390/ijerph17072572_

Round 1
Reviewer 1 Report
I read with great interest the Manuscript titled “Maternal Exposure to Ambient Air Pollution and Pregnancy Complications in Victoria, Australia” (ijerph-734480), which falls within the aim of International Journal of Environmental Research and Public Health. In my honest opinion, the topic is interesting enough to attract the readers’ attention. The methodology is accurate and the conclusions are supported by the data analysis. Nevertheless, the authors should clarify some points and improve the discussion citing relevant and novel key articles about the topic.
Authors should consider the following recommendations:
- Recent and novel evidence suggested that epigenetic changes, in particular altered expression of selective miRNA, may play a key role in both placental-induced diseases such as pre-eclampsia and intrauterine growth restriction. In this regard, I suggest mentioning these recent studies about the topic: PMID: 28466013; PMID: 23354729.
- I would suggest the potential role dioxin and dioxin-like pollutants exposure during early life in the pathogenesis of endometriosis and uterine malformation, which are known to be risk factors for pregnancy complications (refer to: PMID: 25920525; PMID: 22789143).
Author Response
I read with great interest the Manuscript titled “Maternal Exposure to Ambient Air Pollution and Pregnancy Complications in Victoria, Australia” (ijerph-734480), which falls within the aim of International Journal of Environmental Research and Public Health. In my honest opinion, the topic is interesting enough to attract the readers’ attention. The methodology is accurate and the conclusions are supported by the data analysis. Nevertheless, the authors should clarify some points and improve the discussion citing relevant and novel key articles about the topic.
Authors should consider the following recommendations:
- Recent and novel evidence suggested that epigenetic changes, in particular altered expression of selective miRNA, may play a key role in both placental-induced diseases such as pre-eclampsia and intrauterine growth restriction. In this regard, I suggest mentioning these recent studies about the topic: PMID: 28466013; PMID: 23354729.
- I would suggest the potential role dioxin and dioxin-like pollutants exposure during early life in the pathogenesis of endometriosis and uterine malformation, which are known to be risk factors for pregnancy complications (refer to: PMID: 25920525; PMID: 22789143).
Response: Thank you for drawing these studies to our attention. We have given these papers consideration and now make reference to them in our discussion (para 4).
Reviewer 2 Report
This study examines the association between maternal PM2.5 and NO2 exposure and GDM, hypertensive disorders of pregnancy and placental abruption, and could potentially contribute to the growing literature on air pollution and pregnancy complications. However, several issues of concern, particularly the exposure assessment methods, should be addressed.
Introduction:
The case for examining the association between air pollutants and these three outcomes in particular is not strong enough. The statement about the impact of air pollution on these outcomes not being well understood can be strengthened with a brief statement on the state of the literature (discussed later in more detail in the discussion section) to emphasize the rationale for this study and its contribution to the literature, particularly for placental abruption where there is an opportunity to discuss the limited evidence.
Methods –study population:
As this is a study of seasonal exposures (air pollution and temperature), the authors should consider making exclusions to avoid fixed cohort bias (Strand, Barnett and Tong 2011).
Methods –outcome assessment:
I would add a section for outcome assessment, separate from the section on study population.
Methods –assignment of exposure:
Of particular concern is the use of annual average of pollutant concentrations for outcomes experienced during pregnancy, as opposed to examining exposures over specific periods of pregnancy (e.g., a preconception period right before pregnancy, trimester averages, or weekly/ monthly averages). This not only washes out any temporal trends in air pollutant exposures over the year, but could also include exposures irrelevant to the pregnancy period.
This choice of an exposure metric is critical to the study question and the interpretation of the results. The authors do not explain their rationale for using this metric and its appropriateness for the outcomes studied. They explain in the discussions section that they were unable to capture short-term exposures or observe spikes in outcomes. However, this would have also been true had pregnancy-specific periods been used instead of annual averages and is common in most studies of air pollutant exposures and pregnancy outcomes with a similar study design. The main concern here is with using the exposure period relevant to the outcomes in question.
Methods –weather and other covariates:
Why is the metric used for temperature, which is estimated over the entire pregnancy period, different from that used for PM2.5 and NO2? Also, on page 4, paragraph 1, the authors say the meteorological characteristics, including temperature, were for each trimester –this is in contradiction the earlier statement that it was measured as a pregnancy average from conception to delivery. Even with this choice of a whole pregnancy average temperature metric, how did the authors determine the date of conception (e.g.: using the date of the last menstrual period or a different method)?
Methods –statistical analysis:
There was no consideration or adjustment for correlations introduced when examining area-level covariates (e.g.: IRSD and remoteness classification).
Results/Tables:
Table 2:
- The group of hypertensive disorders here does not include eclapmpsia, even though this was included in the outcome definition in the methods section.
- If the results presented are from co-pollutant models, as indicated in the text, the table title or footnote should indicate this.
- The table title should include relevant study information (e.g., person, time and place).
Reference:
Strand LB, Barnett AG, Tong S. Methodological challenges when estimating the effects of season and seasonal exposures on birth outcomes. BMC Med Res Methodol. 2011;11:49.
Author Response
This study examines the association between maternal PM2.5 and NO2 exposure and GDM, hypertensive disorders of pregnancy and placental abruption, and could potentially contribute to the growing literature on air pollution and pregnancy complications. However, several issues of concern, particularly the exposure assessment methods, should be addressed.
Introduction:
The case for examining the association between air pollutants and these three outcomes in particular is not strong enough. The statement about the impact of air pollution on these outcomes not being well understood can be strengthened with a brief statement on the state of the literature (discussed later in more detail in the discussion section) to emphasize the rationale for this study and its contribution to the literature, particularly for placental abruption where there is an opportunity to discuss the limited evidence.
Response: Thank you, we have incorporated more of a discussion about previous studies in the introduction to outline the current state of evidence and added additional wording to rationalise why we studied these outcomes (final paragraph of the introduction).
Methods –study population:
As this is a study of seasonal exposures (air pollution and temperature), the authors should consider making exclusions to avoid fixed cohort bias (Strand, Barnett and Tong 2011).
Response: Thank you for bringing our attention to this. Following your suggestion, we have considered fixed cohort bias and performed an additional sensitivity analysis to address this. These results are presented in Supplementary Table 7. The methods and results have new text that outline this additional analysis.
Methods –outcome assessment:
I would add a section for outcome assessment, separate from the section on study population.
Response: thank you, we have done this (added as 2.1.2)
Methods –assignment of exposure:
Of particular concern is the use of annual average of pollutant concentrations for outcomes experienced during pregnancy, as opposed to examining exposures over specific periods of pregnancy (e.g., a preconception period right before pregnancy, trimester averages, or weekly/ monthly averages). This not only washes out any temporal trends in air pollutant exposures over the year, but could also include exposures irrelevant to the pregnancy period.
This choice of an exposure metric is critical to the study question and the interpretation of the results. The authors do not explain their rationale for using this metric and its appropriateness for the outcomes studied. They explain in the discussions section that they were unable to capture short-term exposures or observe spikes in outcomes. However, this would have also been true had pregnancy-specific periods been used instead of annual averages and is common in most studies of air pollutant exposures and pregnancy outcomes with a similar study design. The main concern here is with using the exposure period relevant to the outcomes in question.
Response: thank you. We agree that the usual of an annual average of exposure is the main limitation of the study and is important to highlight as a key limitation. We have added text in the discussion to highlight the rationale for the use of this exposure; we used an annual average of exposure as it was the only temporal exposure metric available to us from the land use regression model. However, it is a geographically resolved estimate; assigned to maternal address at the level of SA1. As such, we have encountered the trade-off of a geographically resolved estimate that is not temporally resolved. We have ensured that throughout the manuscript, especially the discussion, that this limitation is discussed at length. We also make reference to this study as ‘exploratory’ for this reason, with the emphasise that further studies with more temporally resolved estimates are needed. We believe we have drawn attention to this limitation throughout the paper; it appears in the abstract, discussion and conclusion.
Methods –weather and other covariates:
Why is the metric used for temperature, which is estimated over the entire pregnancy period, different from that used for PM2.5 and NO2? Also, on page 4, paragraph 1, the authors say the meteorological characteristics, including temperature, were for each trimester –this is in contradiction the earlier statement that it was measured as a pregnancy average from conception to delivery. Even with this choice of a whole pregnancy average temperature metric, how did the authors determine the date of conception (e.g.: using the date of the last menstrual period or a different method)?
Response: Thank you. The mention of trimester-specific ambient temperature on page 4 is an error and has been corrected to reflect a whole of pregnancy average. We have added text under ‘weather and other covariates’ to clarify that the estimated date of conception was calculated by subtracting 266 days from the estimated date of confinement (as recorded in the administrative dataset).
Methods –statistical analysis:
There was no consideration or adjustment for correlations introduced when examining area-level covariates (e.g.: IRSD and remoteness classification).
Response: We did carefully consider collinearity in the selection of covariates in the adjusted models. For socioeconomic status for example, IRSD was chosen as it was considered superior to other correlated indicators such as maternal Indigenous status or remoteness, as it is a composite of multiple factors. Remoteness on the other hand was used in the sensitivity analysis only, as a proxy of possible health service access (although noting that it is an imperfect indicator of this).
Results/Tables:
Table 2:
- The group of hypertensive disorders here does not include eclapmpsia, even though this was included in the outcome definition in the methods section.
- Response: Thank you for highlighting this. The omission of eclampsia in the table text was an error and has been corrected.
- If the results presented are from co-pollutant models, as indicated in the text, the table title or footnote should indicate this.
- Response: Footnote added to reflect co-pollutant model.
- The table title should include relevant study information (e.g., person, time and place).
- Response: added, thank you.
Reference:
Strand LB, Barnett AG, Tong S. Methodological challenges when estimating the effects of season and seasonal exposures on birth outcomes. BMC Med Res Methodol. 2011;11:49.
Round 2
Reviewer 2 Report
The authors have addressed most of my comments and I am satisfied with the revisions made to parts of the manuscript. I have some minor (and one major) comments:
MINOR COMMENTS
Methods:
There may be an error in how the fixed cohort bias exclusions were conducted or reported. I believe the Strand, Barnett and Tong method was to exclude pregnancies with a conception date that precedes the study start date by the length of the shortest pregnancy (in this study, assuming 22 weeks) or precedes the end date by the length of the longest pregnancy (assuming 42 weeks). In other words, including those with a conception start date less than 22 weeks before the start date and more than 42 weeks before the end date. If, as currently reported in the manuscript, births with conception dates less than 42 weeks of the study end date are included, those with longer pregnancies might give birth after the study end date of Dec. 31, 2015 and are missed. While this is not likely to change the results of the sensitivity analysis, it should be corrected.
Results:
Table 2: one further suggestion –indicating that the results are from co-pollutant models is a major descriptor of the results presented and should be included in the table title or in the existing footnote.
MAJOR COMMENTS
Methods and Discussion:
My main concern regarding the use of the exposure metric remains. While I agree the limitations section on this particular point is extensive, I still think it does not address the issue of relevant time periods. Perhaps the authors can strengthen this choice of a metric by including evidence, if any exist, on why it is biologically plausible/relevant.
Also, pregnancy exposures, which most of the extant literature has examined and for which there is suggestive evidence of an association with these outcomes of interest, are not accounted for in this study, making it difficult conclusions about the annual average as an exposure of interest without also examining other relevant periods.
Finally, this choice of a metric makes it harder to draw comparisons, including to a previous cohort study by the same team in which trimester exposures were used. This fact should at least be noted in the discussion section before comparisons are made to previous findings.
Author Response
The authors have addressed most of my comments and I am satisfied with the revisions made to parts of the manuscript. I have some minor (and one major) comments:
MINOR COMMENTS
Methods:
There may be an error in how the fixed cohort bias exclusions were conducted or reported. I believe the Strand, Barnett and Tong method was to exclude pregnancies with a conception date that precedes the study start date by the length of the shortest pregnancy (in this study, assuming 22 weeks) or precedes the end date by the length of the longest pregnancy (assuming 42 weeks). In other words, including those with a conception start date less than 22 weeks before the start date and more than 42 weeks before the end date. If, as currently reported in the manuscript, births with conception dates less than 42 weeks of the study end date are included, those with longer pregnancies might give birth after the study end date of Dec. 31, 2015 and are missed. While this is not likely to change the results of the sensitivity analysis, it should be corrected.
Response: Thank you for bringing our attention to this. Your correct in that we made an error in defining the cohort. We have corrected this now to include pregnancies with an estimated conception date <22 weeks before the cohort start date and >42 weeks before the cohort end date. We have repeated the analysis and corrected text in the methods and Supplementary Table 7 to reflect this. As you suspected, the correction did not meaningfully alter the results presented in Supplementary Table 7.
Results:
Table 2: one further suggestion –indicating that the results are from co-pollutant models is a major descriptor of the results presented and should be included in the table title or in the existing footnote.
Response: this is currently stated in the footnote for Table 2 (appears as ‘Footnote: results for co-pollutant models presented.’)
MAJOR COMMENTS
Methods and Discussion:
My main concern regarding the use of the exposure metric remains. While I agree the limitations section on this particular point is extensive, I still think it does not address the issue of relevant time periods. Perhaps the authors can strengthen this choice of a metric by including evidence, if any exist, on why it is biologically plausible/relevant.
Response: Thank you for highlighting these concerns. We have expanded discussion in the limitations paragraph (discussion para 5) to discuss your points regarding the annual exposure metric further. This includes a discussion of the usual exposure periods studied in perinatal epidemiology studies, other studies that utilise an annual average and how the use of our annual metric limits comparability. We also highlight why the annual metric is relevant (as a surrogate for a woman’s long-term background exposure).
Also, pregnancy exposures, which most of the extant literature has examined and for which there is suggestive evidence of an association with these outcomes of interest, are not accounted for in this study, making it difficult conclusions about the annual average as an exposure of interest without also examining other relevant periods.
Response: we aren’t entirely sure we understand this comment. Is the reviewer referring to other environmental exposures in pregnancy? Or perhaps the importance of understanding whether there are potentially important periods of exposure during pregnancy and these outcomes?
If it is the latter, we agree that understanding the importance of pre-conception, versus trimester one, two or three exposures is important and we feel this has been raised in the discussion. As we have reflected that the use of an annual metric of exposure is most comparable to studies utilising whole of pregnancy exposures, we have made mention of this in the discussion. Hopefully our expanded discussion in para 5 addresses this concern.
Finally, this choice of a metric makes it harder to draw comparisons, including to a previous cohort study by the same team in which trimester exposures were used. This fact should at least be noted in the discussion section before comparisons are made to previous findings.
Response: We have ensured that we have drawn attention to this in comparing to other studies, with additional text to reflect this in para 2 and 3 of the discussion. We haven’t drawn attention to our previous study that you’ve mentioned, primarily because it was exploring a severe smoke event, rather than ambient air pollution.